# TF-SCORE: TIME-SERIES FORECASTING USING SCORE-BASED DIFFUSION MODEL

## ABSTRACT

Diffusion models have emerged as powerful generative models, capable of synthesizing high-quality images by capturing complex underlying patterns. Building on this success, these models have been adapted for time-series forecasting, a domain characterized by intricate temporal dependencies. However, most existing works have focused primarily on empirical performance without sufficient theoretical exploration. In this paper, we address this gap by introducing a generalized loss function within the diffusion-based forecasting framework. Leveraging this foundation, we introduce TF-score, a score-based diffusion model designed to capture the interdependencies between historical data and future predictions. Extensive experiments across six benchmark datasets show that TF-score consistently surpasses leading baselines, including prior diffusion-based models. Furthermore, we extend existing guidance sampling strategies into a our score-based formulation, achieving performance gains across multiple datasets while providing a detailed analysis of the trade-offs involved.

## 1 INTRODUCTION

Time-series data is prevalent in our daily lives. There are numerous sub-problem related to time-series data such as time-series generation, forecasting, anomaly detection and sequential recommendation (Ahmed et al., 2010; Ismail Fawaz et al., 2019; Fu, 2011). Among these, time-series forecasting is a prominent example, aiming to predict future behavior based on historical data. Despite extensive research efforts to develop effective forecasting methods, the inherent complexity of time-series data presents significant challenges. To solve these challenge, researchers have increasingly turned to deep learning techniques to better understand and model the structure of time-series data (Lim & Zohren, 2021; Torres et al., 2021; Miller et al., 2024).

Generative Models has achieved remarkable success across various real-world applications. Notably, these methods have been employed to generate high-quality synthetic images, realistic voices, and even lifelike videos, among other creative outputs. Recently, the conditional generation strategies of generative models have widely conducted to time-series processing, including time-series generation, forecasting and anomaly detection.

Diffusion models, which have recently gained attention for their ability to generate high-quality images while maintaining stable training loss, have also been applied to time-series processing, achieving state-of-the-art results. However, many of these applications have lacked theoretical justification, focusing instead on the empirical adaptation of diffusion models to new domains (Rasul et al., 2021; Tashiro et al., 2021; Yan et al., 2021).

In this work, we examine the application of diffusion models to time-series data, classifying existing approaches into two main categories and analyzing their target values from the perspective of the score function (c.f. Section 2.2). We further generalize these categories into a continuous score stochastic differential equation (score SDE) framework, addressing the theoretical gaps in previous studies. Building on this, we introduce TF-score, a score-based diffusion model for time-series forecasting that captures the internal structure of historical and predicted values by optimizing a generalized loss function. This formulation leads to a more robust denoising process, improving predictive performance. Leveraging modifications to a well-known backbone model (Kong et al., 2021), TF-score surpasses most of the baselines, demonstrating superior performance compared to other diffusion-based forecasting methods.

Additionally, we extend existing guidance strategies used in diffusion models into a score-based form, exploring their performance in time-series forecasting (Ho & Salimans, 2022; Kollovieh et al., 2023). This can be achieved through our specific choice of generation process: synthesizing a total sequence. By generating entire time-series, TF-score enables comparison between the generated output and historical data, facilitating more robust guidance sampling. Through a series of experiments, we demonstrate that these approaches enhances model performance for several datasets while shows trade-off relationship across datasets.

Our contributions can be summarized as follows:

- We unify various diffusion models into a more generalized framework by deriving the continuous score SDE form (c.f. Section 3).

- We apply previous guidance generation methods for diffusion models to our score-based framework and evaluate their performance (c.f. Section 4).

- As a result, our model, TF-score, achieves state-of-the-art results on six of the most popular forecasting datasets, supported by extensive ablation studies.

## 2 PRELIMINARY AND PROBLEM STATEMENT

### 2.1 DIFFUSION MODELS

Generative models aim to synthesize realistic data, such as images, by learning the underlying probability distribution of the data (Oussidi & Elhassouny, 2018; Harshvardhan et al., 2020; Cao et al., 2024). Among various generative approaches, diffusion models have gained prominence defeating generative adversarial network (GAN), in terms of generating high-quality images with more stable training (Dhariwal & Nichol, 2021; Song et al., 2020; Ho et al., 2020; Cao et al., 2024). Diffusion models operate through following two-step process: i) **Noising** step, which means gradually adding noise to an image, transforming it into Gaussian noise, ii) **Denoising** step, which means recovering the original image from the noisy version, where the noise is sampled from a specific distribution, typically a normal distribution (Yang et al., 2023).

Initially, the denoising process was designed to reverse the noising process by adding noise in the opposite direction at each step. This process is derived from minimizing the Kullback-Leibler (KL) divergence between the joint probability of noising and denoising step, leading to an inequality involving the negative log-likelihood, similar to the variational autoencoder (VAE) framework. This approach is called Denoising Diffusion Probabilistic Models (DDPMs) (Ho et al., 2020). Below, we briefly describe the DDPM process.

Given original image $\mathbf{x} \sim p(\mathbf{x})$ and the length of noising and denoising step $T$, DDPMs add noise to the image according to the transition kernel: $p(\mathbf{x}_t|\mathbf{x}_{t-1}) = \mathcal{N}(\mathbf{x}_t; \sqrt{1-\beta_t}\mathbf{x}_{t-1}, \beta_t\mathbf{I})$, where $t \in \{1, 2, ..., T\}$ and $\beta_t \in (0, 1)$ is a hyperparameter. With sufficiently large $T$, $\mathbf{x}_t$ converges to a normal distribution. DDPMs then train a corresponding learnable denoising kernel $p_\theta(\mathbf{x}_{t-1}|\mathbf{x}_t) = \mathcal{N}(\mathbf{x}_{t-1}; \mu_\theta(t, \mathbf{x}_t), \Sigma(t, \mathbf{x}_t))$, where the denoising process aims to reverse the added noise. By minimizing the KL divergence between the joint distributions of the noising and denoising processes, DDPMs optimize target network $\epsilon_\theta(\cdot, \cdot)$ by using the following loss function:

$$L_{DDPM}(\theta) = \mathbb{E}_t \mathbb{E}_{\mathbf{x} \sim p(\mathbf{x})} \mathbb{E}_{\epsilon \sim \mathcal{N}(\mathbf{0}, \mathbf{I})} [\lambda(t) ||\epsilon - \epsilon_\theta(t, \mathbf{x}_t)||^2]$$

As a follow-up research, Song et al. (2020) have generalized diffusion models from discrete-time processes to continuous Stochastic Differential Equation (SDE) formulations, introducing Variance Exploding (VE), Variance Preserving (VP), and sub-VP processes. In this framework, the noising and denoising processes of diffusion models are reinterpreted as forward and reverse SDEs, respectively:

$$d\mathbf{x} = \mathbf{f}(t, \mathbf{x})dt + g(t)d\mathbf{w}$$

$$d\mathbf{x} = [\mathbf{f}(t, \mathbf{x}) - g(t)^2 \nabla_\mathbf{x} \log p(t, \mathbf{x})]dt + g(t)d\bar{\mathbf{w}}$$

, where $t \in [0, 1]$, $f$ is an affine and $\mathbf{w}$, $\bar{\mathbf{w}}$ represent forward and backward Brownian motion, respectively. Among these, the VP process is particularly notable for its connection to DDPMs, where: $\mathbf{f}(t, \mathbf{x}) = -\frac{1}{2}\beta(t)\mathbf{x}$, $g(t) = \sqrt{\beta(t)}$. They demonstrate that diffusion models train score network $s_\theta(\cdot, \cdot)$ to learn a gradient of log likelihood, score function, by using following score matching loss:

$$L_{SM}(\theta) = \mathbb{E}_t \mathbb{E}_{\mathbf{x}_t \sim p(\mathbf{x}_t)}[\lambda(t)||s_\theta(t, \mathbf{x}_t) - \nabla_{\mathbf{x}_t}\log p(\mathbf{x}_t)||^2]$$

However, directly using score matching loss is computationally prohibitive since calculating exact score function of $\mathbf{x}_t$ needs statistical method (Hyvärinen, 2005; Song et al., 2020). Thanks to specific formulation of $\mathbf{f}$ and $g$, we can derive a following denoising score matching loss, which can be calculated by using given formula (Vincent, 2011; Øksendal, 2014):

$$L_{DSM}(\theta) = \mathbb{E}_t \mathbb{E}_{\mathbf{x} \sim p(\mathbf{x})} \mathbb{E}_{\mathbf{x}_t \sim p(\mathbf{x}_t|\mathbf{x})}[\lambda(t)||s_\theta(t, \mathbf{x}_t) - \nabla_{\mathbf{x}_t}\log p(\mathbf{x}_t|\mathbf{x})||^2]$$

We can directly derive the equivalence between $L_{DDPM}(\theta)$ and $L_{DSM}(\theta)$ by considering the structure of the forward SDE. The drift term $\mathbf{f}(\cdot, \cdot)$ is affine and the diffusion term $g(\cdot)$ depends solely on the diffusion step. This results in the conditional probability $p(\mathbf{x}_t|\mathbf{x})$ being represented as a Gaussian distribution, $\mathcal{N}(\mathbf{x}_t; \mu_t(\mathbf{x}), \sigma_t)$ (Øksendal, 2014). Therefore, we can compute the gradient of log likelihood, $\nabla_{\mathbf{x}_t}\log p(\mathbf{x}_t|\mathbf{x})$, as: $\nabla_{\mathbf{x}_t}\log p(\mathbf{x}_t|\mathbf{x}) = -(\mathbf{x}_t - \mathbf{x})/\sigma_t^2 = -\epsilon/\sigma_t$, where the reparametrization trick is used on $\mathbf{x}_t = \mu_t(\mathbf{x}) + \sigma_t\epsilon$ and $\epsilon \sim \mathcal{N}(\mathbf{0}, \mathbf{I})$. Then by parameterizing the score network $s_\theta(t, \mathbf{x}_t) = -\epsilon_\theta(t, \mathbf{x}_t)/\sigma_t$, we can show that $L_{DDPM}(\theta) \sim L_{DSM}(\theta) \sim L_{SM}(\theta)$. Specifically, note that $\epsilon_\theta(t, \mathbf{x}_t) = -\sigma_t s_\theta(t, \mathbf{x}_t) \sim -\sigma_t \nabla_{\mathbf{x}_t}\log p(\mathbf{x}_t)$, which will be used in our description of classifier-free guidance (CFG) (Ho & Salimans, 2022) in Section 4.1.

Once the score network is trained, diffusion models proceed with the denoising step. At this stage, there are two main sampling strategies: the predictor-corrector (PC) sampler and a deterministic sampler based on the probability flow ordinary differential equation (ODE). In here, we explain PC sampler that is used in our experiment. The PC sampler works by first estimating the next step using a known numerical SDE solver, which is called *predictor*. Then refining the estimate with a score-based MCMC strategy, which is named of *corrector*. A representative example of predictor is an Euler-Maruyama sampling predictor, which is a discretization of backward SDE:

$$\mathbf{x}_{t-1} = [\mathbf{f}(t, \mathbf{x}_t) - g(t)^2 s_\theta(t, \mathbf{x}_t)]\Delta t + g(t)\Delta w, \qquad t \in [1, 0] \quad \text{and} \quad \Delta w \sim \mathcal{N}(\mathbf{0}, \Delta t\mathbf{I})$$

, where $\Delta t$ is a time interval.

Song et al. (2020) achieved state-of-the-art results through extensive hyperparameter tuning of various SDEs, predictors and correctors. However, for our experiments, we adopt the VP SDE and use an Euler-Maruyama sampling predictor without corrector, which is a default setting of it (Song et al., 2020). This allows us to isolate the performance of TF-score from other factors, ensuring that other control variables remain fixed.

## 2.2 TIME-SERIES FORECASTING

Time-series forecasting involves predicting future values based on historical data (Lim & Zohren, 2021; Torres et al., 2021; Miller et al., 2024). Specifically, given a historical sequence $\mathbf{x}^{1:N}$, the task is to forecast the future sequence $\mathbf{x}^{N+1:N+T}$, where $N$ represents the length of the historical data, and $T$ represents the length of the prediction. Each data point $\mathbf{x}$ belongs to $\mathbb{R}^d$. For clarity, we define $\mathbf{x}^{hist}$ by a sequence of history, $\mathbf{x}^{1:N}$, $\mathbf{x}^{pred}$ by a future values, $\mathbf{x}^{N+1:N+T}$, and $\mathbf{x}^{total}$, a total sequence $\mathbf{x}^{1:N+T}$. Time-series forecasting has been widely researched improve the accuracy of future predictions. However, the complex, intertwined characteristics of time-series data make it difficult to fully capture and understand its underlying patterns.

To address this challenge, researchers have increasingly turned to generative models, which aim to model the conditional likelihood of time-series data and provide a more comprehensive understanding of its structure. As a result, many time-series diffusion models were appeared, which can generally be classified into two categories: those that generate only future values ($\mathbf{x}^{pred}$) (Rasul et al., 2021; Tashiro et al., 2021) and those that generate the entire sequence ($\mathbf{x}^{total}$) (Lim et al.,

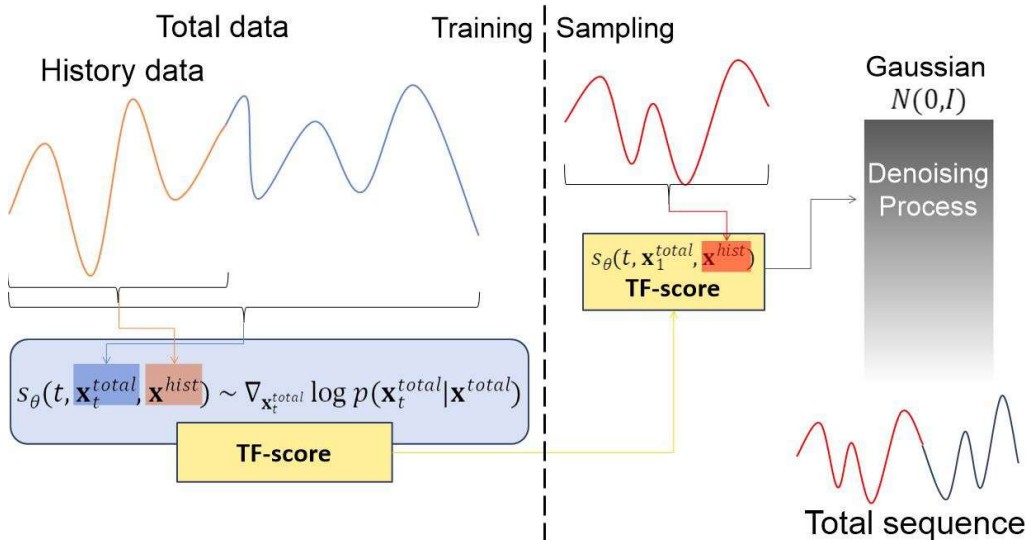

Figure 1: Overall framework of TF-score.

2023; Kollovieh et al., 2023; Lim et al., 2024). We provide a detailed explanation of their contributions and the rationale behind their target selection in Section 6. Therefore, those who apply DDPM methods to forecasting problem optimize one of the following equations:

$$L_{DDPM}^{pred}(\theta) = \mathbb{E}_t \mathbb{E}_{\mathbf{x}^{pred}} \mathbb{E}_{\epsilon \sim \mathcal{N}(\mathbf{0}, \mathbf{I})}[\lambda(t) || \epsilon - \epsilon_\theta(t, \mathbf{x}_t^{pred}, \mathbf{x}^{hist}) ||^2] \tag{1}$$

$$L_{DDPM}^{total}(\theta) = \mathbb{E}_t \mathbb{E}_{\mathbf{x}^{total}} \mathbb{E}_{\epsilon \sim \mathcal{N}(\mathbf{0}, \mathbf{I})}[\lambda(t) || \epsilon - \epsilon_\theta(t, \mathbf{x}_t^{total}, \mathbf{x}^{hist}) ||^2] \tag{2}$$

Based on our review, most existing forecasting methods leveraging diffusion models are derived from DDPMs, and thus, the majority of diffusion-based forecasting approaches optimize one of these loss functions.

## 3 GENERALIZATION OF EXISTING METHODS

In this section, we generalize two types of DDPM-based methods (Eq. 1 and 2) to a continuous score SDE formulation and establish the connection between these two diffusion model approaches.

### 3.1 ANALYSIS OF EXISTING METHOD

In Section 2.2, we introduced two representative diffusion-based forecasting models, which optimize $L_{DDPM}^{pred}$ and $L_{DDPM}^{total}$. From a score-matching perspective, these models can be interpreted as optimizing a score network $s_\theta(\cdot, \cdot, \cdot)$ through the following score-matching losses (c.f. Section 2.1):

$$L_{SM}^{pred}(\theta) = \mathbb{E}_t \mathbb{E}_{\mathbf{x}_t^{pred}}[|| s_\theta(t, \mathbf{x}_t^{pred}, \mathbf{x}^{hist}) - \nabla_{\mathbf{x}_t^{pred}} \log p(\mathbf{x}_t^{pred}|\mathbf{x}^{hist}) ||_2^2]$$

$$L_{SM}^{total}(\theta) = \mathbb{E}_t \mathbb{E}_{\mathbf{x}_t^{total}}[|| s_\theta(t, \mathbf{x}_t^{total}, \mathbf{x}^{hist}) - \nabla_{\mathbf{x}_t^{total}} \log p(\mathbf{x}_t^{total}|\mathbf{x}^{hist}) ||_2^2]$$

, where $t \in [0, 1]$. Note that we ignore weights for computational convenience.

Although these equations optimize the conditional score function to generate target sequences, directly computing them is computationally prohibitive due to the need for statistical methods (Hyvärinen, 2005; Song et al., 2020). Therefore, we derive the denoising score-matching losses to train the score network $s_\theta$:

**Theorem 1** *For each $L_{SM}^{pred}(\theta)$ and $L_{SM}^{total}(\theta)$, its denoising score matching are represented as follows:*

$$L_{DSM}^{pred}(\theta) = \mathbb{E}_t \mathbb{E}_{\boldsymbol{x}^{total}} \mathbb{E}_{\boldsymbol{x}_t^{total}}[||s_\theta(t, \boldsymbol{x}_t^{pred}, \boldsymbol{x}^{hist}) - \nabla_{\boldsymbol{x}_t^{pred}} logp(\boldsymbol{x}_t^{total}|\boldsymbol{x}^{total})||_2^2] \qquad (3)$$

$$L_{DSM}^{total}(\theta) = \mathbb{E}_t \mathbb{E}_{\boldsymbol{x}^{total}} \mathbb{E}_{\boldsymbol{x}_t^{total}}[||s_\theta(t, \boldsymbol{x}_t^{total}, \boldsymbol{x}^{hist}) - \nabla_{\boldsymbol{x}_t^{total}} logp(\boldsymbol{x}_t^{total}|\boldsymbol{x}^{total})||_2^2] \qquad (4)$$

*Therefore, these models aim same conditional score function since $\nabla_{\boldsymbol{x}_t^{total}} logp(\boldsymbol{x}_t^{total}|\boldsymbol{x}^{total}) = \nabla_{[\boldsymbol{x}_t^{hist}, \boldsymbol{x}_t^{pred}]} logp(\boldsymbol{x}_t^{total}|\boldsymbol{x}^{total})$.*

A proof of Theorem 1 is provided in Appendix A. Based on Theorem 1, we demonstrate that previous methods have essentially optimized the same underlying model. The key difference lies in the input; $L_{DSM}^{total}(\theta)$ considers the diffused value of the entire sequence and this distinction introduces significant advantages. Specifically, considering diffused conditions holistically leads to performance improvements (see Table 1) and enables the use of guidance sampling, which will be discussed in Section 4. This is because the denoising process considers both the historical context and future predictions simultaneously while the score network $s_\theta(t, \mathbf{x}_t^{total}, \mathbf{x}^{hist})$ captures the

Table 1: Comparison of $CRPS_{sum}$ results between $L_{DSM}^{pred}$ and $L_{DSM}^{total}$ on Exchange and Electricity datasets.

|  | Exchange | Electricity |
|---|---|---|
| $L_{DSM}^{pred}$ | .013±.000 | .032±.000 |
| $L_{DSM}^{total}$ | **.006±.000** | **.017±.000** |

internal structure of total sequence, leading to a more meaningful denoising process. Therefore, we suggest to use $L_{DSM}^{total}$ for diffusion-based forecasting and propose TF-score, which optimize score function for time-series forecasting.

Beyond Equation 4, we place additional emphasis on the prediction portion of the sequence. In designing TF-score, we aim to ensure that it generates a predictive sequence that takes past history into account but is not overly dominated by historical values. To achieve this balance, we introduce a hyperparameter $\gamma$ to control the influence of the past history, which is fixed by 0.1. The exact loss function is then defined as:

$$l(\theta) = ||s_\theta(t, \mathbf{x}_t^{total}, \mathbf{x}^{hist}) - \nabla_{\mathbf{x}_t^{total}} logp(\mathbf{x}_t^{total}|\mathbf{x}^{total})||^2,$$
$$L(\theta) = \mathbb{E}_t \mathbb{E}_{\mathbf{x}^{total}} \mathbb{E}_{\mathbf{x}_t^{total}}[||\gamma\mathbf{m} \otimes l(\theta) + (1-\mathbf{m}) \otimes l(\theta)||_1]$$

, where $\otimes$ is a hadamard product and $\mathbf{m} = \{x_{ij}\}_{(N+T)\times d}$ is a mask vector that $x_{ij} = 1$ if $i \leq N$ and 0 otherwise, dividing the past and future elements in our loss function.

## 3.2 EXPERIMENTS

In this section, we present the results of several experiments conducted to evaluate the performance of TF-score. We outline the experimental setups and discuss the outcomes.

### 3.2.1 EXPERIMENTAL SETUPS

We first describe the diffusion architecture used in TF-score. To effectively capture the conditional score function along the temporal axis, we adapt DiffWave (Kong et al., 2021) to our settings. Since TF-score is based on DiffWave, we highlight the key differences. As derived in Theorem 1, the input consists of the diffusion timestep, the diffused target data, and historical data, i.e. $t, \mathbf{x}^{hist}, \mathbf{x}_t^{total}$. Consistent with previous works (Ho et al., 2020; Kong et al., 2021), the timestep t is embedded into a continuous domain using sinusoidal embedding:

$$embbeding(t) = [\sin(t/N^{0/d}), ..., \sin(t/N^{d-1/d}), \cos(t/N^{0/d}), ..., \cos(t/N^{d-1/d})]$$

, where d is embedding dimension and N is hyperparameterset to 128 and 10,000, respectively. Furthermore, since we generalize Equation 1, 2 to score SDE, we use VP SDE and an Euler-Maruyama

sampling predictor without corrector, which are a generalized formulation of DDPM (c.f. Section 2.1) and a default setting of VP SDE in Song et al. (2020), respectively.

For evaluation, we use the sum of continuous ranked probability score ($CRPS_{sum}$), a widely recognized metric for probabilistic forecasting. CRPS measures the compatibility between the cumulative distribution function (CDF) $F$ and an observation $x$ as $CRPS(F, x) = \int (F(z) - \mathbb{I}(x \leq z))^2 dz$, where $\mathbb{I}$ is an indicator function. To approximate CDF, we use an empirically estimated CDF $\hat{F} = \frac{1}{N} \sum_{i=1}^{N} \mathbb{I}(x_i \leq z)$, where $x_i$ are samples from $F$. Then we compute the sum of CRPS over all features, denoted as $CRPS_{sum}$,

$$CRPS_{sum}(F, x) = \frac{CRPS(F, \sum_i x_{i,t})}{\sum_{i,t} |x_{i,t}|}$$

, where $\sum_{i,t} |x_{i,t}|$ means the summation of all target features at time $t$.

Next, we use TF-score on 6 widely-used time-series forecasting datasets: Exchange (Lai et al., 2017), Solar (Lai et al., 2017), Electricity[1], Traffic[2], Taxi[3], Wikipedia[4]. We give detailed description of these datasets in Table 5, including dimension, total number of timesteps, domain and frequency data of each dataset. We also report hyperparameters setting in Table 5: the history and prediction lengths, the number of diffusion steps, and the number of iterations. Here, we point out that we follow the common practice of training based on iteration count and saving checkpoints every 5,000 steps, as done in other diffusion models (Ho et al., 2020; Song et al., 2020).

After training TF-score on the selected datasets, we evaluate its performance against a wide range of baseline models. These baselines include: i) classical multivariate methods such as VAR, VAR-Lasso (Lütkepohl, 2005), GARCH (van der Weide, 2002), and VES (Hyndman et al., 2008); ii) RNN-based methods like Vec-LSTM-ind-scaling, Vec-LSTM-lowrank-Copula, GP-scaling, and GP-Copula (Salinas et al., 2019); iii) Transformer-based models, specifically Transformer-MAF (Rasul et al., 2020); and iv) VAE and diffusion-based models, including KVAE (Fraccaro et al., 2017), TimeGrad (Rasul et al., 2021), and CSDI (Tashiro et al., 2021). A detailed description of these baseline models can be found in Appendix C.

### 3.2.2 EXPERIMENTAL RESULTS

We present the $CRPS_{sum}$ performance of TF-score and other baseline models in Table 4. We evaluate TF-score with 5 different seeds, and we report both the mean and standard deviation. As shown in the table, TF-score consistently outperforms all competing models across every dataset, including other diffusion-based forecasting models. Notably, while diffusion-based forecasting models like TimeGrad and CSDI perform comparably on certain datasets, TF-score consistently delivers superior results across a wide range of data complexities, from relatively low-dimensional datasets (e.g., Exchange) to high-dimensional ones (e.g., Wiki).

## 4 APPLICATIONS OF GUIDANCE SAMPLING

In this section, we explore several guidance methods — classifier-free guidance (CFG) (Ho & Salimans, 2022), replacement method (Song et al., 2020; Ho et al., 2022), and observation self guidance (OSG) (Kollovieh et al., 2023) — and their application within our score-based diffusion framework.

### 4.1 CLASSIFIER FREE GUIDANCE

To incorporate an auxiliary classifier in naïve conditional generation, Dhariwal & Nichol (2021) introduced classifier guidance, modifying the standard denoising process by adjusting the estimated noise. Originally, $\epsilon(\mathbf{x}_t|\mathbf{c}) \sim -\sigma_t \nabla_{\mathbf{x}_t} \log p(\mathbf{x}_t|\mathbf{c})$ is replaced with $\tilde{\epsilon}(\mathbf{x}_t|\mathbf{c}) = \epsilon(\mathbf{x}_t|\mathbf{c}) -$

---

[1]https://archive.ics.uci.edu/ml/datasets/ElectricityLoadDiagrams20112014

[2]https://archive.ics.uci.edu/ml/datasets/PEMS-SF

[3]https://www1.nyc.gov/site/tlc/about/tlc-trip-record-data.page

[4]https://github.com/mbohlkeschneider/gluon-ts/tree/mv_release/datasets

Table 2: $CRPS_{sum}$ results on evaluation datasets. The best scores are in boldface.

| | Exchange | Solar | Electricity | Traffic | Taxi | Wiki |
|---|---|---|---|---|---|---|
| VES | .005±.000 | .900±.003 | .880±.004 | .350±.002 | - | - |
| VAR | .005±.000 | .830±.006 | .039±.001 | .290±.005 | - | - |
| VAR-Lasso | .012±.000 | .510±.006 | .025±.000 | .150±.002 | - | 3.10±.004 |
| GARCH | .023±.000 | .880±.002 | .190±.001 | .370±.002 | - | - |
| KVAE | .014±.002 | .340±.025 | .051±.019 | .100±.005 | - | .095±.012 |
| Vec-LSTM ind-scaling | .008±.001 | .391±.017 | .025±.001 | .087±.041 | .506±.005 | .133±.002 |
| Vec-LSTM low-copula | .007±.000 | .319±.011 | .064±.008 | .103±.006 | .326±.007 | .241±.033 |
| GP scaling | .009±.000 | .368±.012 | .022±.000 | .079±.000 | .183±.395 | 1.48±1.03 |
| GP copula | .007±.000 | .337±.024 | .025±.002 | .078±.002 | .208±.183 | .086±.004 |
| Transformer MAF | .005±.003 | .301±.014 | .021±.000 | .056±.001 | .179±.002 | .063±.003 |
| TimeGrad | .006±.001 | .287±.020 | .021±.001 | .044±.006 | .114±.020 | .049±.002 |
| CSDI | .007±.001 | .298±.004 | .017±.000 | .020±.001 | .123±.003 | .047±.003 |
| TF-score | **.005±.000** | **.224±.008** | **.017±.000** | **.020±.000** | **.113±.001** | **.046±.001** |

$w\sigma_t\nabla_{\mathbf{x}_t}\log p(\mathbf{c}|\mathbf{x}_t)$, where $w$ is a weighting term, and an additional classifier is trained to calculate $p(\mathbf{c}|\mathbf{x}t)$. From the perspective of score-based SDEs, this approach can be interpreted as altering the score function $\nabla_{\mathbf{x}_t}\log p(\mathbf{x}_t|\mathbf{c})$ to $\nabla_{\mathbf{x}_t}\log\tilde{p}(\mathbf{x}_t|\mathbf{c}) = \nabla_{\mathbf{x}_t}\log p(\mathbf{x}_t|\mathbf{c}) + w\nabla_{\mathbf{x}_t}\log p(\mathbf{c}|\mathbf{x}_t) = \nabla_{\mathbf{x}_t}\log p(\mathbf{x}_t|\mathbf{c})p(\mathbf{c}|\mathbf{x}_t)^w$, which means $\tilde{p}(\mathbf{x}_t|\mathbf{c}) \sim p(\mathbf{x}_t|\mathbf{c})p(\mathbf{c}|\mathbf{x}_t)^w$ and effectively incorporating the classifier into the generative process.

To address the dependency on an additional classifier, Ho & Salimans (2022) proposed classifier-free guidance (**CFG**), allowing the generation process to be guided without the need for a separately trained classifier. In CFG, the model learns the modified noise estimate $\tilde{\epsilon}(\mathbf{x}_t|\mathbf{c}) = (1 + w)\epsilon(\mathbf{x}_t|\mathbf{c}) - w\epsilon(\mathbf{x}_t)$ by training a single model that handles both conditional and unconditional generations. This is achieved by training with zero-padding for the unconditional case, resulting in $\tilde{\epsilon}_\theta(\mathbf{x}_t, \mathbf{c}) = (1 + w)\epsilon_\theta(\mathbf{x}_t, \mathbf{c}) - w\epsilon_\theta(\mathbf{x}_t, \mathbf{0})$.

From a score matching perspective, this can be understood as $\nabla_{\mathbf{x}_t}\log\tilde{p}(\mathbf{x}_t|\mathbf{c}) = \nabla_{\mathbf{x}_t}\log p(\mathbf{x}_t|\mathbf{c}) + w\nabla_{\mathbf{x}_t}\log p(\mathbf{c}|\mathbf{x}_t) = \nabla_{\mathbf{x}_t}\log p(\mathbf{x}_t|\mathbf{c}) + w\nabla_{\mathbf{x}_t}(\log p(\mathbf{x}_t|\mathbf{c}) - \log p(\mathbf{x}_t)) = (1 + w)\nabla_{\mathbf{x}_t}\log p(\mathbf{x}_t|\mathbf{c}) - w\nabla_{\mathbf{x}_t}\log p(\mathbf{x}_t)$. And this formulation leads to the generalized score function used in CFG: $\tilde{s}_\theta(\mathbf{x}_t, \mathbf{c}) = (1 + w)s_\theta(\mathbf{x}_t, \mathbf{c}) - ws_\theta(\mathbf{x}_t, \mathbf{0})$, where $\mathbf{0}$ means zero padding. We use this generalized CFG sampling. As the formulation shows, CFG should train both conditional and unconditional sampling to single model. In line with Ho & Salimans (2022), we adopt a proportional training strategy, where with probability $p_{\text{cond}}$, the model trains the conditional score network $s_\theta(\mathbf{x}_t, \mathbf{c})$, and with probability $1 - p_{\text{cond}}$, it trains the unconditional score network $s_\theta(\mathbf{x}_t, \mathbf{0})$.

### 4.2 OBSERVATION SELF GUIDANCE

Traditional forecasting models typically employ an architecture where the historical data ($\mathbf{x}^{hist}$) is provided as input, and the model outputs future predictions ($\mathbf{x}^{pred}$). However, if we design diffusion models to the entire time-series sequence ($\mathbf{x}^{total} = [\mathbf{x}^{hist}, \mathbf{x}^{pred}]$), the generation process can be continuously guided by the history.

Previous works, such as Song et al. (2020) and Ho et al. (2022), suggest generating the entire sequence $\mathbf{x}^{total}$ by continually replacing the historical part during the diffusion process. More specifically, for a diffusion model $p_\theta(\mathbf{x}_t^{total}|\mathbf{x}^{hist})$ at a denoising step $t$, the history part of $\mathbf{x}_t^{total}$ is sequentially replaced with the corresponding diffused historical values from the forward SDE, $\mathbf{x}_t^{hist}$. We denote this approach as the **replacement method**.

However, naïvely applying replacement method cannot give meaningful results. This may occur because the forward SDE applied to $\mathbf{x}^{hist}$ is not fully aligned with the backward SDE used for denoising $\mathbf{x}_t^{total}$, mainly due to the stochastic nature of the Brownian motion driving both processes. There-

fore, alternative strategies must be considered, such as observation self guidance (**OSG**) proposed by Kollovieh et al. (2023). Similar to controllable generation of (Song et al., 2020), Kollovieh et al. (2023) starts from bayes' rule: $p_\theta(\mathbf{x}_t^{total}|\mathbf{x}^{hist}) \sim p_\theta(\mathbf{x}^{hist}|\mathbf{x}_t^{total})p_\theta(\mathbf{x}_t^{total}|\mathbf{x}^{hist})$, which means

$$\nabla_{\mathbf{x}_t^{total}}\log p_\theta(\mathbf{x}_t^{total}|\mathbf{x}^{hist}) \sim \nabla_{\mathbf{x}_t^{total}}\log p_\theta(\mathbf{x}^{hist}|\mathbf{x}_t^{total}) + \nabla_{\mathbf{x}_t^{total}}\log p_\theta(\mathbf{x}_t^{total}|\mathbf{x}^{hist}).$$

Thus, the posterior score function $\nabla_{\mathbf{x}^{total}t}\log p_\theta(\mathbf{x}^{hist}|\mathbf{x}_t^{total})$ enables more accurate sampling in the diffusion process. To estimate this posterior, Kollovieh et al. (2023) assumes that the posterior distribution follows a multivariate Gaussian distribution (refer to Kollovieh et al. (2023) for other assumption): $p_\theta(\mathbf{x}^{hist}|\mathbf{x}_t^{total}) = \mathcal{N}(\mathbf{x}^{hist}; \hat{\mathbf{x}}^{hist}, \mathbf{I})$, where $\hat{\mathbf{x}}^{hist}$ represents the restored historical values and can be computed as follows (adapted from Song et al. (2022)): $\hat{\mathbf{x}}^{total} = (\mathbf{x}_t^{total} - \sqrt{1-\bar{\alpha}_t}\epsilon_\theta(t, \mathbf{x}_t^{total}, \mathbf{x}^{hist}))/\sqrt{\bar{\alpha}_t}$, where $\alpha_t = 1-\beta_t, \bar{\alpha}_t = \prod_{i=1}^t \alpha_i$. This leads to the posterior score function $\nabla_{\mathbf{x}_t^{total}}\log p_\theta(\mathbf{x}^{hist}|\mathbf{x}_t^{total})$ being expressed as the mean-squared error (MSE) between $\mathbf{x}^{hist}$ and $\hat{\mathbf{x}}^{hist}$.

Building upon this idea, we incorporate the posterior score correction into TF-score. Inspired by the energy-based sampling approach in Kollovieh et al. (2023), we propose a modified score function:

$$\tilde{s}_\theta(t, \mathbf{x}_t^{total}, \mathbf{x}^{hist}) = s_\theta(t, \mathbf{x}_t^{total}, \mathbf{x}^{hist}) - w\nabla_{\mathbf{x}^{total}}||\hat{\mathbf{x}}^{hist} - \mathbf{x}^{hist}||_2^2$$

, where $w$ is a weighting term that controls the influence of the correction. This approach ensures that the model better aligns the historical part of the sequence during the denoising process.

### 4.3 EXPERIMENTAL SETTINGS AND RESULTS

In this section, we present the experimental results of TF-score using three different guidance sampling strategies: classifier-free guidance (CFG), the replacement method, and observation self-guidance (OSG). For each strategy, we vary the weight parameter $w$ between 0.01 and 0.1, and evaluate the model's performance using the $CRPS_{sum}$ metric on several representative datasets. All experiments are conducted with five different random seeds to compute the mean and standard deviation for robustness.

Especially on OSG, due to its computational complexity which requires calculating the gradient of the mean-squared error (MSE), we were unable to perform experiments using this method on larger datasets like Electricity and Traffic. It is important to note that OSG in Kollovieh et al. (2023) is designed for univariate sequences, where the MSE gradient calculation is feasible. In contrast, our experiments deal with multivariate time-series, which significantly increases the computational cost for OSG.

The experimental results are summarized in Table 3. As shown in the table, OSG and replacement method demonstrate some improvements on the Solar dataset, while CFG performs better on Electricity and Traffic datasets. However, each method has its own limitations: OSG incurs a high computational cost, especially for multivariate time-series, and replacement method is the simplest method that makes inferior results on the other baselines, whereas CFG requires a modified training procedure. This aligns with the "No Free Lunch" (NFL) theorem, highlighting that no single method is optimal for all cases, and trade-offs are inevitable.

Table 3: Results of each guidance sampling.

|  | Original | CFG$_{0.01}$ | CFG$_{0.1}$ | OSG$_{0.01}$ | OSG$_{0.1}$ | Replacement |
|---|---|---|---|---|---|---|
| Exchange | **.0054**±**.0002** | .0059±.0005 | .0057±.0001 | .0055±.0003 | .0062±.0003 | .0057±.0001 |
| Solar | .2241±.0080 | .3121±.0031 | .2997±.0020 | .2230±.0094 | .2270±.0120 | **.2211**±**.0081** |
| Electricity | .0168±.0003 | .0166±.0002 | **.0163**±**.0004** | - | - | .0173±.0005 |
| Traffic | .0202±.0004 | **.0195**±**.0001** | .0199±.0002 | - | - | .0223±.0006 |

## 5 ABLATION EXPERIMENTS

In this section, we present ablation studies conducted across several datasets to analyze the impact of varying the diffusion steps in TF-score. We experiment with different numbers of diffusion steps: 50, 100, 200, 250, 500, and report the corresponding $CRPS_{sum}$ results.

As indicated by the results, there are optimal "sweet spots" for the number of steps depending on the dataset. For example, TF-score requires relatively fewer diffusion steps on datasets like Exchange and Electricity, whereas it benefits from higher steps on the Solar dataset to achieve the best performance. However, since lots of diffusion steps increase sampling time of TF-score, we compromise them by hyperparameters in Table 5 in Appendix B.

We also point out that an notable distinction of TF-score, compared to other diffusion-based forecasting models such as CSDI (Tashiro et al., 2021) and TimeGrad (Rasul et al., 2021), is its ability to adjust the number of sampling steps without the need for additional training at each specific step. This flexibility offers a significant advantage, as it allows TF-score to adapt more efficiently across varying datasets and conditions, without incurring extra computational costs for retraining.

Table 4: Results of ablation study varying the number of sampling steps

|  | 50 | 100 | 200 | 250 | 500 |
|---|---|---|---|---|---|
| Exchange | 0057±.0003 | **.0054±.0002** | .0057±.0002 | .0059±.0004 | .0057±.0002 |
| Electricity | .0168±.0003 | **.0165±.0005** | .0168±.0007 | .0166±.0005 | .0166±.0002 |
| Solar | .4540±.0125 | .2829±.0090 | .2241±.0080 | .2313±.0059 | **.2155±.0089** |

## 6 RELATED WORKS

In this section, we review diffusion-based time-series models, focusing on their use of score functions and categorizing them based on their target score objectives.

As introduced in Sections 2.2 and 3, existing diffusion-based models for time-series forecasting can be broadly divided into two categories: those that learn the score function $\nabla_{\mathbf{x}_t^{pred}} \log p(\mathbf{x}_t^{pred}|\mathbf{x}^{hist})$, which focuses solely on the predicted sequence, and those that learn the score $\nabla \mathbf{x}_t^{total} \log p(\mathbf{x}_t^{total}|\mathbf{x}^{hist})$, which models the entire sequence. Below, we discuss these categories in detail.

Models targeting $\nabla_{\mathbf{x}_t^{pred}} \log p(\mathbf{x}_t^{pred}|\mathbf{x}^{hist})$ focus on generating the predicted sequence from the history. Two prominent models in this category are TimeGrad (Rasul et al., 2021) and CSDI (Tashiro et al., 2021). TimeGrad, a well-known diffusion-based forecasting model, generates predictions autoregressively. Given a historical sequence $\mathbf{x}^{hist} = \mathbf{x}^{1:N}$, it generates the next value $\mathbf{x}^{N+1}$. Then, using the sequence $\mathbf{x}^{2:N+1}$, it generates $\mathbf{x}^{N+2}$, recursively continuing this process to achieve $\mathbf{x}^{pred} = \mathbf{x}^{N+1:N+T}$. In this setup, TimeGrad can be seen as learning $\nabla_{\mathbf{x}_t^{pred}} \log p(\mathbf{x}_t^{pred}|\mathbf{x}^{hist})$ for single-step prediction. CSDI (Tashiro et al., 2021), on the other hand, generates the entire prediction sequence $\mathbf{x}^{pred}$ in one shot, given the historical data $\mathbf{x}^{hist}$. While one-shot generation may appear more efficient than autoregressive methods, it can introduce higher variance in the generated samples due to the randomness of the backward SDE. To mitigate this, CSDI evaluates the model by averaging results over 100 generated sequences to ensure stability.

Next, we explain models targeting $\nabla_{\mathbf{x}_t^{total}} \log p(\mathbf{x}_t^{total}|\mathbf{x}^{hist})$. Kollovieh et al. (2023) focus on generating the entire sequence, including both historical and predicted values, with the goal of improving conditional generation through history-guided sampling. TSDiff (Kollovieh et al., 2023) introduces this approach, generating the complete sequence $\mathbf{x}^{total}$ and guiding the generation process using the history sequence $\mathbf{x}^{hist}$. The guidance mechanisms employed in these models are discussed in detail in Section 4.

Although designed for time-series generation, Lim et al. (2023) and Lim et al. (2024) take a different approach by generating the full time-series autoregressively within a latent space, specifically

for handling irregularly sampled time-series data. This latent space generation allows for better modeling of complex time dependencies, offering an alternative to standard methods. Therefore, researchers have focused on total generation to take account of additional techniques, such as guidance sampling or generation in latent space.

Our proposed method, TF-score, can be seen as a unified framework that leverages the strengths of both categories: it generates high-quality predictions while incorporating guidance sampling mechanisms to enhance performance and flexibility. By combining both aspects, TF-score provides a more comprehensive approach to time-series forecasting with diffusion models.

## 7 CONCLUSION

We presented a score-based forecasting model, TF-score, for generalized diffusion framework. Through the proposed methods, our method considers both the historical context and future predictions simultaneously and thereby captures the internal structure of total sequence, leading to a more robust denoising process. We also extend existing guidance strategies used in diffusion models into a score-based form, exploring their performance in time-series forecasting. Through a series of experiments, we demonstrate that these approaches enhance model performance for several datasets while shows trade-off relationship across datasets.

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

## A    DETAILED PROOF

In this section, we give detailed proof of Theorem 1.

We prove denoising score matching loss of prediction, $L_{DSM}^{total}(\theta)$. The result of $L_{DSM}^{pred}(\theta)$ can be derived similarly. We start from decomposing it:

$$L_{SM}^{total}(\theta) = -2 \cdot \mathbb{E}_t \mathbb{E}_{\mathbf{x}_t^{total}} \langle s_\theta(t, \mathbf{x}_t^{total}, \mathbf{x}^{hist}), \nabla_{\mathbf{x}_t^{total}} \mathrm{log} p(\mathbf{x}_t^{total}|\mathbf{x}^{hist}) \rangle$$
$$+ \mathbb{E}_t \mathbb{E}_{\mathbf{x}_t^{total}} \left[ \left\| s_\theta(t, \mathbf{x}_t^{total}, \mathbf{x}^{hist}) \right\|_2^2 \right] + C_1$$

Here, $C_1$ is a constant that does not depend on the parameter $\theta$, and $\langle \cdot, \cdot \rangle$ means the inner product. Then, the first part's expectation of the right-hand side can be expressed as follows:

$$\mathbb{E}_t \mathbb{E}_{\mathbf{x}_t^{total}} \langle s_\theta(t, \mathbf{x}_t^{total}, \mathbf{x}^{hist}), \nabla_{\mathbf{x}_t^{total}} \mathrm{log} p(\mathbf{x}_t^{total}|\mathbf{x}^{hist}) \rangle$$

$$= \int_{\mathbf{x}_t^{total}} \langle s_\theta(t, \mathbf{x}_t^{total}, \mathbf{x}^{hist}), \nabla_{\mathbf{x}_t^{total}} \mathrm{log} p(\mathbf{x}_t^{total}|\mathbf{x}^{hist}) \rangle \mathrm{p}(\mathbf{x}_t^{total}|\mathbf{x}^{hist}) d\mathbf{x}_t^{total}$$

$$= \int_{\mathbf{x}_t^{total}} \langle s_\theta(t, \mathbf{x}_t^{total}, \mathbf{x}^{hist}), \frac{1}{\mathrm{p}(\mathbf{x}^{hist})} \frac{\partial \mathrm{p}(\mathbf{x}_t^{total}, \mathbf{x}^{hist})}{\partial \mathbf{x}_t^{total}} \rangle d\mathbf{x}_t^{total}$$

$$= \int_{\mathbf{x}^{total}} \int_{\mathbf{x}_t^{total}} \langle s_\theta(t, \mathbf{x}_t^{total}, \mathbf{x}^{hist}), \frac{1}{\mathrm{p}(\mathbf{x}^{hist})} \frac{\partial \mathrm{p}(\mathbf{x}_t^{total}, \mathbf{x}^{hist}, \mathbf{x}^{total})}{\partial \mathbf{x}_t^{total}} \rangle d\mathbf{x}_t^{total} d\mathbf{x}^{total}$$

$$= \int_{\mathbf{x}^{total}} \int_{\mathbf{x}_t^{total}} \langle s_\theta(t, \mathbf{x}_t^{total}, \mathbf{x}^{hist}), \frac{\partial \mathrm{p}(\mathbf{x}_t^{total}|\mathbf{x}^{total})}{\partial \mathbf{x}_t^{total}} \rangle \frac{\mathrm{p}(\mathbf{x}^{hist}, \mathbf{x}^{total})}{\mathbf{x}^{hist}} d\mathbf{x}_t^{total} d\mathbf{x}^{total}$$

$$= \int_{\mathbf{x}^{total}} \int_{\mathbf{x}_t^{total}} \langle s_\theta(t, \mathbf{x}_t^{total}, \mathbf{x}^{hist}), \frac{\partial \mathrm{p}(\mathbf{x}_t^{total}|\mathbf{x}^{total})}{\partial \mathbf{x}_t^{total}} \rangle \mathrm{p}(\mathbf{x}^{total}|\mathbf{x}^{hist}) d\mathbf{x}_t^{total} d\mathbf{x}^{total}$$

$$= \mathbb{E}_{\mathbf{x}^{total}} \left[ \int_{\mathbf{x}_t^{total}} \langle s_\theta(t, \mathbf{x}_t^{total}, \mathbf{x}^{hist}), \frac{\partial \mathrm{p}(\mathbf{x}_t^{total}|\mathbf{x}^{total})}{\partial \mathbf{x}_t^{total}} \rangle d\mathbf{x}_t^{total} \right]$$

$$= \mathbb{E}_{\mathbf{x}^{total}} \left[ \int_{\mathbf{x}_t^{total}} \langle s_\theta(t, \mathbf{x}_t^{total}, \mathbf{x}^{hist}), \nabla_{\mathbf{x}_t^{total}} \log p(\mathbf{x}_t^{total}|\mathbf{x}^{total}) \rangle \mathrm{p}(\mathbf{x}_t^{total}|\mathbf{x}^{total}) d\mathbf{x}_t^{total} \right]$$

$$= \mathbb{E}_{\mathbf{x}^{total}} \mathbb{E}_{\mathbf{x}_t^{total}} [\langle s_\theta(t, \mathbf{x}_t^{total}, \mathbf{x}^{hist}), \nabla_{\mathbf{x}_t^{total}} \log p(\mathbf{x}_t^{total}|\mathbf{x}^{total}) \rangle]$$

$$= \mathbb{E}_{\mathbf{x}^{total}} \mathbb{E}_{\mathbf{x}_t^{total}} [\langle s_\theta(t, \mathbf{x}_t^{total}, \mathbf{x}^{hist}), \nabla_{\mathbf{x}_t^{total}} \log p(\mathbf{x}_t^{total}|\mathbf{x}^{total}) \rangle]$$

The second part's expectation of the right-hand side can be rewritten similarly, therefore we can derive following result:

$$L_{SM}^{total}(\theta) = -2 \cdot \mathbb{E}_t \mathbb{E}_{\mathbf{x}^{total}} \mathbb{E}_{\mathbf{x}_t^{total}} \langle s_\theta(t, \mathbf{x}_t^{total}, \mathbf{x}^{hist}), \nabla_{\mathbf{x}_t^{total}} \mathrm{log} p(\mathbf{x}_t^{total}|\mathbf{x}^{hist}) \rangle$$
$$+ \mathbb{E}_t \mathbb{E}_{\mathbf{x}^{total}} \mathbb{E}_{\mathbf{x}_t^{total}} \left[ \left\| s_\theta(t, \mathbf{x}_t^{total}, \mathbf{x}^{hist}) \right\|_2^2 \right] + C_1$$
$$= L_{DSM}^{total}(\theta) + C_1$$

$C$ is a constant that does not depend on the parameter $\theta$.

## B    DESCRIPTIONS OF DATASETS AND HYPERPARAMETERS

In this section, we describe detailed explanations about datasets and hyperparameters.

Table 5: Description of datasets and hyperparameters.

|  | Dimension | Timesteps | Domain | $L_{hist}$ | $L_{pred}$ | $N_{step}$ | $N_{iter}$ /5000 |
|---|---|---|---|---|---|---|---|
| Exchange | 8 | 6071 | $\mathbb{R}^+$ | 90 | 30 | 100 | 36 |
| Solar | 137 | 7009 | $\mathbb{R}^+$ | 72 | 24 | 200 | 67 |
| Electricity | 370 | 5833 | $\mathbb{R}^+$ | 72 | 24 | 50 | 53 |
| Traffic | 963 | 4001 | (0,1) | 72 | 24 | 50 | 31 |
| Taxi | 1214 | 1488 | $\mathbb{N}$ | 48 | 24 | 50 | 17 |
| Wiki | 2000 | 792 | $\mathbb{N}$ | 90 | 30 | 250 | 6 |

## C   DETAILED EXPLANATIONS ABOUT BASELINES

In this section, we describe brief explanation about baselines.

- VAR : a multivariate linear auroregressive model
- VAR-Lasso : VAR regularized by Lasso
- GARCH : a multivariate heteroskedastic model
- VES : a sort of state space model
- KVAE : a variational autoencoder (VAE) to describe dyamics of data
- Vec-LSTM : connect dynamics of input and gaussian distribution of output by using RNN
- Transformer-MAF : transformer-based forecasting model by conditioning temporal dynamics and masked autoregressiveness flow
- TimeGrad : a representative diffusion-based forecasting model
- CSDI : a representative diffusion-based imputation model, which can be applied to forecasting task