# OpenReview forum: "TF-score: Time-series Forecasting using score-based diffusion model"
_ICLR.cc/2025/Conference — ICLR 2025 Conference Withdrawn Submission_

### Official Review · Reviewer_Swz1 · 2024-10-23

**Soundness:** 2
**Presentation:** 1
**Contribution:** 1
**Rating:** 3
**Confidence:** 4

**Summary:**

This paper introduces TF-score, a score-based diffusion model for time series forecasting with specific design in capturing the interdependencies between historical data and future predictions. The authors state the relationship between DDPM-based time series forecasting model and score-matching based methods and unify the framework by deriving the continuous score SDE form. Extensive experiments are conducted to show better performance against multiple baseline models.

**Strengths:**

- The proposed conditional score matching view of time series forecasting loss function is valid, and the idea of weighing errors for history and future generations is insightful.
- The experiments are extensive, covering multiple time series forecasting baselines, multiple guidance sampling methods, and various diffusion step settings, showing efficacy of the proposed model.

**Weaknesses:**

- The similar idea of using score matching model seems to have been proposed by an earlier work [1], so the novelty and contribution of this paper may be quite limited.
- The experiment part largely follows the experiment in [1], and the most advanced time series diffusion models are missing for comparison. To name just a few, [2][3][4][5]
- Applying guidance sampling application on the proposed method cannot prove the superiority against baselines, since similar modifications are not applied on baseline models. It's unclear whether the proposed method can outperform baseline methods in these settings.

[1] Yan, Tijin, et al. "Scoregrad: Multivariate probabilistic time series forecasting with continuous energy-based generative models." https://arxiv.org/abs/2106.10121v1.
[2] Li, Yan, et al. "Generative time series forecasting with diffusion, denoise, and disentanglement." Advances in Neural Information Processing Systems 35 (2022): 23009-23022.
[3] Shen, Lifeng, and James Kwok. "Non-autoregressive conditional diffusion models for time series prediction." International Conference on Machine Learning. PMLR, 2023.
[4] Fan, Xinyao, et al. "MG-TSD: Multi-Granularity Time Series Diffusion Models with Guided Learning Process." The Twelfth International Conference on Learning Representations. 2024.
[5] Ashok, Arjun, et al. "TACTiS-2: Better, Faster, Simpler Attentional Copulas for Multivariate Time Series." The Twelfth International Conference on Learning Representations. 2024.

**Questions:**

- What is the key point that makes the proposed method different from [1]?
- Could you provide a more thorough comparison against the newer baselines to validate the proposed model?
- What can the diffusion-based baseline methods gain from applying guidance sampling method? Can the proposed method still outperform them in those cases?

---

### Official Review · Reviewer_WEys · 2024-10-26

**Soundness:** 2
**Presentation:** 2
**Contribution:** 2
**Rating:** 3
**Confidence:** 4

**Summary:**

This work builds on a growing literature of time series forecasting methods that use diffusion models for forecasting. First, the authors propose an analysis of families of related work, to introduce the forecasting problem and to cluster methods based on their loss function formulation. Here it is argued that generating the entire sequence (history + future values) has advantages over autoregressive approaches that use the history to predict future values.

The authors propose TF-score, a method that is supposed to generalize existing methods, and that can also incorporate a guidance mechanism. Experiments on well-known datasets indicate that the proposed method is on par with and sometimes superior to alternatives from the state of the art, including two diffusion-based methods. An ablation study complements the experiments, indicating that proper configuration of the proposed method is dataset dependent.

**Strengths:**

* I think the proposed objective of this paper is important, because in principle it could subsume existing diffusion-based time-series modeling approaches by presenting a more general formulation
* Experiments and ablation studies are thorough, by considering several datasets and several alternatives from the literature, not limited to diffusion-based approaches to forecasting.
* Despite some typos and a mathematical notation that can be improved, the article reads well and is easy to follow

**Weaknesses:**

* Sec.2.1 is very informal, and more importantly, uses precious space to outline methods that are not directly used in this work (only to review existing work in Sec. 2.2). Since score-based diffusion models through the lenses of stochastic differential equations are well known today, I suggest to make this part more compact, but at the same time more formal. Please, also double check grammar/typos (e.g., line 108: $f$ is an affine and $w$, …) and mathematical notation (check the commas at the end of the equations, e.g. line 108, line 141). Concerning the informal tone: see the comment below $L_{SM}$, the score term there is just not analytically available, therefore Song et al. condition the score on the initial sample, such that it becomes accessible, resulting in $L_{DSM}$. It is not a matter of computational effort or the need for statistical methods.

* Sec. 2.2: pay attention to notation overload: $T$ is used both to indicate diffusion time as well as forecast horizon. Check expression clarity: for example, lines 155,157 do not read well and are vague.
In my opinion, eq.1 and eq.2 are not clear. For eq.1, the input to the score network is $x^{\text{hist}}$, as well as the nosy $x^{\text{pred}}$. Does it mean that $x^{\text{hist}}$ is not used as a conditioning signal, which would be a natural choice? For eq. 2, the input is $x^{\text{hist}}$ and $x^{\text{total}}$: since $x^{\text{hist}} \in x^{\text{total}}$, couldn’t we see this as an “inpainting" approach?
Since the goal of this section, also according to what claimed in the introduction, is to review and categorize in two classes existing approaches, I think it would be useful to provide the reader with more insights than just the two loss functions.

* Sec. 3: this part contains, to the best of my knowledge, several mathematical mistakes. The first expressions in Sec. 3.1 do not make sense to me, especially regarding the score term which is manipulated without care, and with arbitrary choice of variables that are not compatible with score-based diffusion formalism. What does it mean to take the gradient with respect to $x^{\text{pred}}$ of the log of the conditional density of $x^{\text{pred}}_t$ given $x^{\text{hist}}$? Are you trying to make the correspondence between $x^{\text{hist}}$ and $x_0$ (that is the clean data) and $x^{\text{pred}}_t$ to a noisy version of the clean data $x_t$? Note the problem: $x_t$ can be obtained in closed form from $x_0$, whereas obtaining $x^{\text{pred}}_t$ from $x^{\text{hist}}$ is exactly the problem you are trying to solve.
Similarly, eq. 3 and eq. 4 are shaky. In Eq. 3, how do you compute the gradient of the score (this time properly defined) with respect to $x^{\text{pred}}$, which you do not have access to? In Eq. 4, isn’t it redundant to provide $x^{\text{hist}}$ as an input to the score network $s(\cdot)$, as it is already contained in $x^{\text{total}}$?
Also, in line 212 it is said that weights are ignored for computational convenience. However, the weight $\lambda$ is very important, as it determines what exactly you are optimizing: for example by setting $\lambda(t)=g(t)^2/2$, minimizing the loss corresponds to maximum likelihood training [1, Sec. 2].
Finally, it is necessary to delve into the details of the masking mechanism discussed in lines 251-252. It is used to discern past from future elements, which zeros out the future. This, in my opinion, is equivalent to the setup I alluded to in Sec.2 comments: essentially you can imagine $x^{\text{total}}$ as an image, of which you zero out a region, leaving you with a portion that corresponds to $x^{\text{hist}}$, which is amenable to an “inpainting” interpretation.

Next, in sec 3.2.1 authors speak about generalizations of existing schemes to make the point that their approach is different from DiffWave. This is too strong of a claim, in my opinion.

Finally, in sec 3.2.2 the authors could have discussed in more detail why the proposed method performs (slightly) better than diffusion-based alternatives such as TimeGrad and CSDI. For example, the initial message about classifying existing methods in two categories, and the intended take home message as to modeling $x^{\text{total}}$ would have been stronger if properly compared and discussed.

[1] Vahdat, Arash and Kreis, Karsten and Kautz, Jan, “Score-based Generative Modeling in Latent Space”, NeurIPS 2021.

* Sec. 4: apart from traditional guidance, in Sec. 4.2 observation self-guidance is discussed (often attributing ideas to work from Song and Ho, which to the best of my knowledge do not refer to time-series data). The expression at line 380, which authors say implement Bayes rule, is ill defined: $p(a|b) \sim p(b|a) p(a|b)$? The authors should double check, and consequently double check expression at line 383.

Overall, I think the authors should do a better job at explaining the conditioning signal used for their guided method. Only the variant presented in the equation at line 397 is well defined. As a side note, it would have helped quite a lot numbering the various equations used throughout this paper.

**Questions:**

* The claim for the superiority of generating $x^{\text{total}}$ should be better supported. As hinted in the notes above (see weaknesses), a better description (in Appendix C) of the two alternative diffusion-based methods is in order, such as to clarify by means of your experiments what is exactly the pain point of auto-regressive generation. The results in Table 1 are not sufficient in my opinion, as there is not enough detail (apart from just indicating a different loss) to discern exactly what is going on. Results in Table 2 for competitor diffusion-based approaches are better than the numbers reported in Table 1, so I would like to understand if these are “toy models” applied to real data to make the point, or if these correspond to full-fledged models from the literature that use different modeling approaches.
* I think a big pass on the mathematical rigor of this work is in order. Can you please double check your expressions and derivations? This includes the proof provided in Appendix A, which to me appears as a sequence of manipulations that do not stand an appropriate derivation of the loss as most of the related work cited by the authors do.
* Can you clarify the role of a guidance mechanism? Unfortunately, the presented experiments in Table 3 give the impression that guidance is only marginally beneficial, and it is hard to find a “one-size fits all” configuration.

---

### Official Review · Reviewer_8BgG · 2024-11-01

**Soundness:** 2
**Presentation:** 2
**Contribution:** 1
**Rating:** 3
**Confidence:** 4

**Summary:**

This paper proposes using a score-based diffusion model for time series forecasting. The main claimed contribution is the simultaneous generation of both historical context and future predictions using diffusion models. Another claimed contribution is the continuous score of the SDE form for each generation.

**Strengths:**

S1. the method details are presented clearly
S2. the proposed method is simple, and it is easy to follow

**Weaknesses:**

W1. The investigation is insufficient. Many related works on time series diffusion models are missing, e.g., [1,2]. Some works have used score-based diffusion models for time series prediction [3].

- [1] NIPS'22 Generative Time Series Forecasting with Diffusion, Denoise, and Disentanglement
- [2] ICML'23 Non-autoregressive conditional diffusion models for time series prediction
- [3] ICLR'24 Interpretable Diffusion for General Time Series Generation


W2. the statement below is not very convincing. Using the historical context as a condition to generate the future part (without generating the past window) can still capture the internal structure of the total sequence.
> "our method considers both the historical context and future predictions simultaneously and thereby captures the internal structure of total sequence,"

In contrast, generating historical context may be limited:
1) bring some unexpected bias to degrade the prediction performance when there is some unrelated historical information;
2) increase computational burden when a long history context is included. In the experiments, the authors only use small window sizes (<100) for evaluations. And there is no analysis of computational efficiency.

Thus, generating the whole time series using a score-based diffusion model is not fully convincing.

**Questions:**

N/A

---

### Official Review · Reviewer_eN2u · 2024-11-03

**Soundness:** 2
**Presentation:** 2
**Contribution:** 1
**Rating:** 3
**Confidence:** 4

**Summary:**

This paper introduces TF-score, a score-based diffusion model designed for time-series forecasting. The model applies conditional generative diffusion modeling to time-series forecasting tasks, using historical data as conditioning factors and the complete data sequence as the generation target. Additionally, a mask vector is incorporated into the loss function to separate historical and future data.

**Strengths:**

​1. **Appropriate Application of Diffusion Models**: The paper effectively adapts diffusion models for time-series forecasting, a relatively novel domain for such models, building on their successes in generative tasks.

​2. **Unified Framework**: The authors integrate existing diffusion models for time-series forecasting into a continuous score-based framework, enhancing the theoretical foundation of their approach.

​3. **Experimental Performance**: TF-score outperforms multiple benchmark models across various datasets, validating its effectiveness in practical forecasting tasks.

**Weaknesses:**

​1. **Excessive Background**: The paper dedicates substantial space to explaining the background of diffusion models, the time-series forecasting task, and how diffusion models are applied in this area. The authors even include descriptions of existing embedding and guidance methods and the computation of the CRPS metric. This leaves limited space to discuss their own contributions, making the paper feel more like a course report than a research paper.

​2. **Lack of Innovation**: The paper appears to mainly apply existing conditional generation diffusion models to time-series forecasting tasks, with limited novelty. The only new element is the mask vector, which seems overly simplistic. Only about one-fifth of a page out of the 10-page main text is dedicated to introducing this new method, and this so-called new method merely adds a weight mask when calculating the sequence loss. Besides, the section that introduces this new method is even titled **ANALYSIS OF EXISTING METHOD**. Therefore, I don’t consider this to be an innovation.

**Questions:**

1. Refer to Weaknesses.

2. Most of the benchmark models compared in the experiments are relatively outdated, and CRPS is the sole evaluation metric, which makes the results less convincing. However, it’s difficult to suggest alternative benchmarks since the authors do not clearly explain their model architecture or what aspects are innovative.

3. I’m puzzled why Section 3.1 is titled **ANALYSIS OF EXISTING METHOD** and Section 3.2 jumps straight to **EXPERIMENTS**. Where is the authors’ **METHOD**?

---

### Note · Authors · 2024-11-28

I have read and agree with the venue's withdrawal policy on behalf of myself and my co-authors.